# Perception, Trust, and Motivation in Consumer Behavior for Organic Food Acquisition: An Exploratory Study

**DOI:** 10.3390/foods14020293

**Published:** 2025-01-17

**Authors:** Elena Moroșan, Violeta Popovici, Ioana Andreea Popescu, Adriana Daraban, Oana Karampelas, Liviu Marian Matac, Monica Licu, Andreea Rusu, Larisa-Marina-Elisabeth Chirigiu, Sinziana Opriţescu, Elena Iuliana Ionita, Alina Saulean, Maria Nitescu

**Affiliations:** 1Department of Clinical Laboratory—Hygiene of Nutrition, Faculty of Pharmacy, “Carol Davila” University of Medicine and Pharmacy, 020956 Bucharest, Romania; elena.morosan@umfcd.ro (E.M.); sinziana.opritescu@drd.umfcd.ro (S.O.); 2Center for Mountain Economics, “Costin C. Kiriţescu” National Institute of Economic Research (INCE-CEMONT), Romanian Academy, 725700 Vatra-Dornei, Romania; 3Rural Development Research Platform, “Gh. Zane” Institute of Economic and Social Research, Romanian Academy, Iași Branch, 700481 Iași, Romania; 4Department of Pharmaceutical Technology and Biopharmacy, Faculty of Pharmacy, “Carol Davila” University of Medicine and Pharmacy, 020956 Bucharest, Romania; andreea-ioana.popescu@umfcd.ro (I.A.P.); oana.karampelas@umfcd.ro (O.K.); 5Faculty of Pharmacy, “Vasile Goldiș” Western University of Arad, 310045 Arad, Romania; rusu.andreea@uvvg.ro; 6Faculty of Accounting And Management Information Systems, University of Economic Studies, 020956 Bucharest, Romania; liviu.matac@cig.ase.ro; 7Department of Medical Psychology, Faculty of Medicine, “Carol Davila” University of Medicine and Pharmacy, 050474 Bucharest, Romania; monica.licu@umfcd.ro; 8Faculty of Pharmacy, University of Medicine and Pharmacy Craiova, Petru Rareș 2, 200349 Craiova, Romania; larisa.chirigiu@umfcv.ro; 9Faculty of Pharmacy, “Carol Davila” University of Medicine and Pharmacy, 020956 Bucharest, Romania; elena.ionita@drd.umfcd.ro (E.I.I.); alina.stirbu@mst.umfcd.ro (A.S.); 10Department of Hygiene and Medical Ecology, Faculty of Medicine, “Carol Davila” University of Medicine and Pharmacy, 050474 Bucharest, Romania; maria.nitescu@umfcd.ro; 11“Prof. Dr. Matei Bals” National Institute of Infectious Disease, “Carol Davila” University of Medicine and Pharmacy, 020956 Bucharest, Romania

**Keywords:** organic food, eco-food certification, eco-food label, health and environmental benefits, perception, trust, motivation, acquisition behavior

## Abstract

(1) Background: A sustainable healthy diet assures human well-being in all life stages, protects environmental resources, and preserves biodiversity. This work investigates the sociodemographic factors, knowledge, trust, and motivations involved in organic food acquisition behavior. (2) Methods: An online survey via Google Forms platform, with 316 respondents, was conducted from 1 March to 31 May 2024. (3) Results: Our findings show that suitably informed people with high educational levels (academic and post-college) report significant satisfaction with organic food consumption (*p* < 0.05). There is also a considerable correlation between ages 25–65, moderate to high satisfaction, and “yes” for eco-food recommendations (*p* < 0.05). The same satisfaction levels are associated with medium confidence in eco-food labels and a moderate to high monthly income (*p* < 0.05). Our results show that monthly income and residence are not essential factors in higher price perception. Insignificant price variation perception correlated with high confidence and weekly acquisition (*p* < 0.05). Similar price perception correlates with the highest confidence level and daily acquisition (*p* < 0.05). Obese respondents exhibited minimal satisfaction and opted for “abstention” from eco-food recommendations (*p* < 0.05). (4) Conclusions: The present study extensively analyzed Romanian people’s knowledge, perception, and trust regarding organic foods. It demonstrates that sociodemographic factors differentiate consumers and influence attitudes and motivation for organic food acquisition.

## 1. Introduction

Lifestyle substantially influences human physical and mental health [1]. The modern lifestyle—defined by unhealthy diets; sedentarism; smoking; alcohol, medications, and other substance abuse [2]; misuse and addiction to various technologies; and neglecting the balance between professional activities, sleep, and recreation—highly correlates with chronic disease burden and mortality worldwide [3,4]. Fortunately, lifestyle is controllable, and rigorously performing healthy measures over time could reverse the harmful effects of risk factors and increase the quality of life. One of the main measures is changing the modern diet—which consists of high-calorie junk foods that are overprocessed, pumped with chemical additives, sugar-loaded, or based on genetically modified organisms (GMOs, plants or animals)—with a healthy one, rich in legumes [5], fruits [6,7], vegetables [8,9], whole grains [10], and unsaturated fats [11,12].

A sustainable healthy diet maintains human well-being in all life stages at the physical, mental, and social levels; it protects environmental resources and preserves biodiversity [13]. In this context, an organic diet based on eco-food consumption is generally considered a healthy option due to the nutritional benefits (higher antioxidants, optimal fat profile, lower pesticide residues, and limited use of artificial sweeteners and genetically modified organisms [GMOs]) [14,15,16,17].

### 1.1. Literature Review

Eco-food has recently gained significant interest, driven by the growing consumer awareness of health, environmental, and ethical concerns. Understanding consumer behavior toward organic food acquisition is essential for businesses, policymakers, and researchers [18,19,20]. Organic food consumption varies significantly between Romania and the Western and Northern countries belonging to the European Union (EU), reflecting economic factors (market size and penetration, accessibility, and distribution), policy support, cultural norms, consumer awareness and motivation, and key barriers (Table 1).

#### 1.1.1. Organic Food Market Size, Distribution, and Policy Support

The organic food market is well-developed in Western and Northern European countries (Denmark, Germany, France, and Austria), which are leaders in organic food consumption. Organic farming accounts for about 9.6% of the European Union’s agricultural land, and the ‘Farm to Fork’ strategy (F2F) aims to achieve 25% organic agriculture by 2030 [23,24]. The EU Green Deal emphasizes expanding organic farming and consumption as part of its sustainability goals [25]. Officially established in 2021 by the European Parliament, 23 September is the EU Organic Day [26]. On this day of celebration, trends in consumer demands are assessed, awareness of organic products in the supply chain is increased, and new targets for the future of organic production in Europe are finally defined. The annual European Organic Awards Winners significantly contributed to the organic value chain [27]. The EU Organic Awards scheme comprises seven categories and eight awards: the best organic farmer (female and male), the best organic city, the best organic region, the best organic bio-district, the best organic small or medium enterprise (SME), the best organic food retailer, and the best organic restaurant [28,29] (Appendix A). Organic products are widely available, supported by strong retail networks and government subsidies. Most EU countries benefit from robust policies, including subsidies for organic farming, educational campaigns, and incentives for retailers to stock organic products [30,31,32]. Countries such as Germany, Austria, France, and Denmark are leaders in the organic market share, with urban consumers driving demand. Policies supporting organic farming, subsidies, and strong retail distribution channels ensure access across urban and rural areas. Rural populations benefit from subsidies for converting to organic agriculture, which can indirectly increase local availability [33,34,35,36,37,38,39,40,41]. Organic products are widely accessible through supermarkets, specialty stores, and online platforms. Many countries integrate organic products into public institutions, such as schools and hospitals, increasing visibility and accessibility [41,42].

Eastern and Southern European countries (Romania, Bulgaria, and Greece) have emerging organic markets with limited penetration in rural areas. Romania’s organic farming covered about 5.1 acres of agricultural land in 2024 [43,44,45]. Rural consumers often equate “traditional” or “natural” products with organic, reducing demand for certified goods. Due to better distribution networks and higher disposable incomes, organic consumption is concentrated in urban centers. Supermarkets dominate the distribution of organic products, but most are imported. Professional stores and farmer’s markets play a minor role, and online sales of organic products are still underdeveloped [46]. Traditional agricultural practices in rural areas often align with organic principles but lack certification, limiting market growth [47]. The “ae” logo, property of the Ministry of Agriculture and Rural Development (MADR) [48], can be used by the operators/groups of operators of ecologically certified, prepackaged products to identify and promote ecologically certified, prepackaged products and guarantee that the products bearing these logos meet the following conditions: (i) they are produced through organic farming in Romania or contain ingredients that come from organic farming in Romania; and (ii) they are certified by a control commission accredited and approved by the MADR [49]. Strong and consistent organic production and labeling regulations can help ensure consumer trust. Many products are recognized under national quality schemes: 732 traditional products, 171 products obtained from consecrated Romanian recipes, and 1319 mountain products [50,51]. In 2023, over 13,000 organic certificates of all ecologically certified producers in Romania were included in the “Register of Agricultural Products and Producers registered in Organic Agriculture”, an independent initiative to promote organic farmers and their products [50]. However, due to budgetary constraints, the funding of organic farming in the CAP 2023–2027 remained at the same level as in 2014 [52].

#### 1.1.2. Organic Food Awareness, Motivation, and Barriers to Organic Food Consumption

Efforts to make organic foods more affordable and accessible to a broader range of consumers and to substantially increase awareness about the benefits of organic food are centered particularly in Western and Northern European countries [22]. Consumers’ trust in eco-food health benefits, environmental sustainability, and ethical considerations increases organic food consumption. Consumers trust eco-food certifications and the EU Organic Logo and often associate organic products with higher quality [53]. Barriers include higher prices and occasional skepticism about certification authenticity. However, subsidies and consumer education have mitigated these issues in many EU countries. Due to increasing consumer awareness and care for environmental safety, organic food consumption is steadily growing. It has already achieved high levels of organic food consumption, with opportunities for growth in Eastern Europe [32].

Limited market penetration and a focus on exports rather than domestic consumption are the main reasons for Romanians’ significantly lower eco-food consumption than the EU average [46]. Motivation is primarily health-driven, with less emphasis on sustainability and ethical aspects [54]. Skepticism toward certifications and labeling is more pronounced, limiting trust in organic products [55]. Key barriers also include the limited availability and high prices of eco-foods [56,57].

### 1.2. Hypotheses

The literature reviewed led to three hypotheses in the present study:oThe concept of organic food is still not correctly perceived due to the lack of suitable information and transparency; it is a reason for diminished trust in eco-food safety and benefits and a key barrier to acquiring eco-food [58,59,60].oRomanian organic food consumers are highly educated, mostly young or middle-aged, and predominantly women who are highly self-care conscious and adhere to a healthy lifestyle [60,61].oThe trust in organic food certification and official labeling leads consumers to purchase foods with higher environmental sustainability, quality, health benefits, and price due to the credibility attributed to the certifying entity [62,63,64,65,66,67,68,69,70].

### 1.3. Behavioral Theories of Eco-Food Acquisition Applied

Three theories could explain organic food acquisition behavior:oThe Health Belief Model (HBM) focuses on personal perceptions of health-affecting factors and the benefits of preventive actions [71];oValue–Belief–Norm (VBN) posits that environmental behaviors, such as purchasing sustainable or organic food, are driven by personal values, beliefs about the consequences of environmental problems, and a sense of moral obligation to act. Individuals who value environmental protection and believe their actions can make a difference are more likely to choose eco-friendly food options [72,73];oThe Theory of Planned Behavior (TPB) means that people’s availability to purchase healthy foods is higher when they believe it is the right thing to do, if they think their social circle approves, and if they have the resources and knowledge to make healthy choices [59,74,75].

### 1.4. The Aim of the Present Study

The present study explores the impact of Romanian people’s perceptions, trust, and motivation on organic food acquisition. It investigates how consumer behavior regarding organic foods is influenced by age, education, income level, and each person’s interest in maintaining health. Moreover, an extensive statistical analysis correlates the knowledge, perception, attitude, trust, and motivation with sociodemographic data on Romanian consumer behavior regarding eco-food acquisition.

## 2. Methods

### 2.1. Online Questionnaire Presentation

The survey involved voluntary participants ≥18 years old residing in Romania. It was approved by the Ethics Committee of the “Carol Davila” University of Medicine and Pharmacy (Document No. 14357, approved on 30 May 2024). The 30 multiple-choice queries in the questionnaire were formulated based on models from previously published studies [16,74,76,77,78]. It was distributed through online platforms between 1 March and 31 May 2024, and the data were collected electronically in a Microsoft 365 Excel v.2024 workbook. Thirty questions were generated electronically on the Google Form platform (Appendix A). The research team members distributed the URL link via email, SMS, or social and professional networks to colleagues, relatives, and personal contacts. Participants were informed about the survey’s aim, the research team involved, and the time required to complete the questionnaire; moreover, they were assured that no email addresses were collected and that the General Data Protection Regulation (GDPR) guarantees the confidentiality of sensitive personal information. Then, they completed and signed the participation agreement and the individual consent form to enable the publication of research results. Over the course of three months, 316 Romanian residents responded to all 30 questions.

The Romanian consumer perceptions and trust in organic food substantially impact the motivation for eco-food consumption and purchasing behavior [34,79,80]. Therefore, the questionnaire was structured into three distinct parts.

The first questions aim to collect the participants’ sociodemographic data (age, education, residence, sex, and body mass index).

The second group of questions analyzes the respondents’ perception and understanding of the eco-food concept and their beliefs and trust in organic foods’ quality and certification, sustainability, environmental impact, and health benefits.

The third part investigates eco-food preferences, the main criteria for acquiring and consuming eco-food products, the acquisition frequency, the verification of organic food shelf-life and ingredients by reading the eco-food label, the satisfaction rate by consuming eco-food, and potential recommendations.

The above groups were analyzed independently and correlated using the tools mentioned in Section 2.3.

### 2.2. Reliability Analysis

The questionnaire was investigated using the Reliability Analysis Internal Model from XLSTAT Life Sciences v.2024.3.0.1423 by Lumivero (Denver, CO, USA) [81]. The Cronbach’s alpha index and Guttman L1–L6 coefficients were calculated.

### 2.3. Data Analysis

Extensive data analysis used different tools in XLSTAT Life Sciences v.2024.3.0.1423 by Lumivero (Denver, CO, USA): descriptive analysis, ANOVA single factor, correlations between variable parameters from each group, and heat maps [82]. Following the descriptive statistics, the variable parameters are displayed as the absolute frequency (number, *n*) and relative frequency (percentage) [83]. Statistical significance was established at *p* < 0.05 [11].

## 3. Results

### 3.1. Reliability Analysis

This analysis is detailed in the Appendix A. The Cronbach’s alpha index value was 0.926, and the Guttman L1–L6 coefficients were 0.895–1.000. The correlation matrix, covariance matrix (Appendix A), and high coefficients reveal the online questionnaire’s substantial reliability and appreciable internal consistency, thus confirming its high quality. The correlation map in Figure 1 shows that all questions are significantly intercorrelated.

### 3.2. Sociodemographic Data of Participants

Sociodemographic data are registered in Table 2.

Of the total participants, 62.97% are female, 37.03% are male, 80.70% of the respondents have urban residences, 19.39% are from rural zones, 42% of respondents are 35–39 years old, and 34.81% are 25–34 years old (*p* < 0.05). The age groups 19–24 and 55–60 have similar percentages (10.13%), while 2.22% are over 65. Over 80% of participants have academic studies (university—48.10% and post-university—32.91%), while 61.71% are employees, 14.87% are entrepreneurs/owners, 8.23% are homeworkers, 6.33% are students, 5.06% are self-employed, 2.53% are pensioners, and 1.27% are unemployed. Around 27.22% have a monthly income in the range of RON 4001–7000, while 24.37% have over RON 10,000; 14.87% have RON 2001–3000, a similar percentage (12.96%) have RON 3001–4000 and RON 7001–10,000 (*p* > 0.05), and 7.59% have under RON 2000 (*p* < 0.05). BMI values show that 42.41% of participants have normal weight, 29.11% are overweight, and similar percentages (14.24%) are obese and underweight (*p* < 0.05).

### 3.3. Eco-Food Perception

This objective was assessed by investigating the participants’ familiarity with eco-food, their perception of quality, their general attitude towards their consumption, trust in the certifications and controls displaying the organic food logo, and the main factors influencing their purchase decision.

#### 3.3.1. Eco-Food Concept Perception and Understanding

Two questions with four choices available, alone or associated, highlighted the most important aspects regarding the respondents’ perception of the eco-food concept (Figure 2A,B).

Most respondents (183, 57.9%) recognize the specific terms BIO, ECO, and Organic as indicators of organic food products, which suggests a high awareness of the official terminology and trust in the regulations associated with these terms (Figure 2A). One hundred fifty-three respondents (48.4%) consider that food from rural households is organic (*p* < 0.05). The EU logo for organic products is less prevalent than the terms BIO, ECO, and Organic (90 vs. 183 respondents, *p* < 0.05). Several respondents (*n*= 51, 16.1%) confuse the terms “Natural” or “100% Natural” with organic food products. Moreover, 4.11% of respondents (*n*= 13) define eco-food using all four items, 6.64% *n* = 21) through three items, and 25.31% (*n*= 80) through two items (*p* < 0.05). Most respondents (63.92%, *n* = 202) selected only one item representing the eco-food concept (*p* < 0.05, Figure 2A).

Limiting pesticide and additive use is perceived as essential, indicating a significant concern for food health and safety (*n*= 251, 79.4%, Figure 2B). Food safety and higher nutritional value (140 vs. 131 respondents, *p* > 0.05) are also essential. Although sustainability and environmental impact are significant for 97 respondents (30.7%), they are less of a priority than the direct impact on consumers’ health. All aspects are essential for 36 respondents (11.39%), while another 55 (17.40%) and 85 (26.89%) opted for three and two significant ones, respectively (*p* < 0.05). The highest number of participants (140, 44.30%) selected only one main item (*p* < 0.05, Figure 2B).

Numerous other aspects were analyzed to investigate the respondents’ knowledge and understanding of eco-food and to assess their opinions about its benefits for human health and the environment.

#### 3.3.2. Perception of Eco-Food Information—Source and Availability

Supermarkets were the primary data source for most respondents (40.19%) about eco-food. Very few participants mentioned organized eco-food expositions and schools as significant places where interested people could find the requested information (1.90% and 0.95%, respectively, *p* < 0.05). Over 45% of participants believe that the current information on eco-food is insufficient (45.57%), while 54.43% consider its availability to be moderate (42.72%) or enough (11.71%). Moreover, only 24.68% of respondents frequently/regularly update their eco-food production and provenance data, vs. 67.41% occasionally/rarely and 7.91% never (*p* < 0.05).

#### 3.3.3. Perception of Eco-Food’s Impact on Environment and Human Health

Most of the respondents confirm the positive influence of eco-food on the environment (91.77%) through Yes/Maybe yes vs. No/Maybe no/I don’t know/Not significantly (8.23%), *p* < 0.05. Similarly, they confirmed that organic foods are healthier than conventional ones (91.14% vs. 8.86%, *p* < 0.05). Numerous participants believe that eco-food has significant/moderate benefits for human health (88.61% vs. 11.39%, *p* < 0.05) compared to conventional ones.

#### 3.3.4. Eco-Food Price Perception vs. Conventional Ones

Over 90% of respondents believe that eco-food prices are higher than conventional ones (93.99%) vs. similar (5.38%) and lower (0.63%), *p* < 0.05.

#### 3.3.5. Trust in Eco-Food Labels and Romanian-Certified Organic Foods

Over 45% of respondents expressed their confidence (C1–C5, C1—minimal, C5—maximal) in an eco-food label as a medium level (C3, 46.84%), while 41.77% have trust in Romanian-certified organic food (*p* > 0.05, Figure 3).

### 3.4. Eco-Food Acquisition Behavior

Most participants revealed organic food acquisition (95.89%), while only 4.11% reported that they “never” purchased eco-foods (*p* < 0.05, Figure 4A). They opt for frequent acquisition (daily and weekly, 48.73%) or rarely (monthly or occasionally, 47.15%, *p* > 0.05, Figure 4A). Our results also report that only 33.54% of respondents always verify the eco-food ingredients, and 23.73% check their shelf-life by reading the label from the package.

Three queries with multiple choices available, alone or associated, investigated the respondents’ preferences, motivations, and eco-food purchasing behavior (Figure 4B–D).

Fruits and vegetables (*n* = 247, 78.16%) and dairy products and eggs (*n* = 208, 65.82%) are the most frequently purchased organic food categories (*p* < 0.05, Figure 4B).

Honey and other healthy foods (*n* = 169, 53.48%), fish and meat (*n* = 109, 34.49%), and novel foods (chia seeds, protein powders, microalgae, noni, acai, etc., *n* = 80, 25.31%) are also important to consumers (*p* < 0.05, Figure 4B). Basic foods (oil, vinegar, flour, sugar, bread) and sweets are purchased less often in the eco version (*n* = 53 (14.48%) and *n* = 27, (7.37%), *p* < 0.05). Figure 4B also indicates that most respondents selected multiple (2–7) items (*n* = 258, 81.64%); only 18.35% (*n* = 58, *p* < 0.05) opted for only one item.

Most respondents prefer eco-food from supermarkets, hypermarkets (*n* = 235, 74.36%), and local markets (*n* = 156 (49.36%), *p* < 0.05). Pharmacies and health food stores are selected by 26.58% of respondents (*n* = 84), neighborhood stores are commonly frequented by 7.91% (*n* = 25), 53 participants (16.77%) opted for online acquisitions, and only 11 (3.48%) indicated other sources (*p* < 0.05, Figure 4C). A total of 178 (56.32%) respondents selected 2-5 items, and 138 (43.67%) marked only one choice (*p* < 0.05).

Price, taste, and odor are the most common criteria for eco-food acquisition (*n* > 200, Figure 4D). They are followed in decreasing order by provider and country (*n* = 176), aspect (*n* = 144), and friend/family recommendations (*n* = 120), while 40 respondents mention eco-friendly packages (*p* < 0.05). The data from Figure 4D also reveal that most participants (*n* = 303) selected multiple choices (2–6), while only 13 opted for one item (*p* < 0.05).

### 3.5. Eco-Food Perception, Trust, and Motivation Influence Eco-Food Acquisition Behavior

Two questions with multiple choices available, alone or associated, investigated the respondents’ motivations for purchasing or avoiding eco-food (Figure 5A,B).

The main reasons for acquiring eco-food are illustrated in Figure 5A. Concern for their health is the main priority for most attendants. With 253 respondents (80.06%), this aspect significantly outperforms other reasons, indicating that consumers strongly emphasize eco-food benefits on health. The following two essential motivations were (i) eco-food being of high quality and (ii) limiting pesticide and additive use in organic food production (*n* = 131 and 128, *p* > 0.05, Figure 5A). It denotes that many consumers perceive eco-food as better quality than conventional food, reducing exposure to harmful chemicals and artificial additives in their daily diet.

The responses of numerous survey participants suggest that there is a tendency to support local farmers by preferring their eco products (*n* = 83), as well as an awareness and desire to reduce the negative impact on the environment *n* = 48) through sustainable food choices (*p* < 0.05, Figure 5A). Multiple choices (2–5) were recorded for 182 respondents (57.59%), while 134 (42.41%), *p* < 0.05, selected only one item.

Substantial motivations to avoid eco-food are displayed in Figure 5B. More than 65% of respondents cite the high prices of organic food products as the main reason. Low availability and lack of trust in eco-food quality are considerable obstacles to purchasing organic food products, according to 119 and 117 respondents, respectively. Thus, limited access to organic food products is an important barrier because of geographical location or reduced store offerings. Additionally, significant suspicion among consumers regarding the authenticity and incontestable quality of the organic products on the market is a considerable obstacle. Another remarkable cause is a lack of information about the benefits (*n* = 37). These highlighted the multiple barriers that prevent consumers from purchasing organic food products, especially the importance of economic and reliability aspects. Around 181 participants marked one item, while 145 opted for 2–4 choices (*p* < 0.05).

### 3.6. Sociodemographic Factors Differentiate the Consumers and Influence Their Trust and Motivation for Eco-Food Acquisition

Many factors influence eco-food consumption behavior, leading to various acquisition frequencies, satisfaction levels (S1–S5; S1—minimal, S5—maximal), and potential eco-food recommendations (Figure 6A,B).

Over 50% of respondents (54.75%) reveal high satisfaction levels (S4, 38.24% and S5, 15.51%) regarding eco-food consumption, 39.24% are moderately satisfied (S3 level), while only 6.01% reported minimal satisfaction rates (S2 and S1 levels, Figure 6A).

The correlation matrix in the Appendix A shows that the higher price perception of eco-food is significantly associated with all incomes (except for RON 2001–3000), rural and urban residence, C2, and S1, S3–S5 (r = 0.898–0.995, *p* < 0.05). No significant (NS) price is correlated with C4, S1, S2, weekly acquisition, and RON 2001–3000 (r = 0.884–0.999, *p* < 0.05). Lower price perception strongly correlates with monthly acquisition and C1 (r = 0.999, *p* < 0.05). Similar price perception substantially correlates with daily acquisition and C5 (r = 0.980–0.999, *p* < 0.05). No significant price perception is remarkably linked with RON 2001–3000/month income, eco-food acquisition weekly, C4, S1, and S2 (r = 0.884–0.999, *p* < 0.05). S3–S5 are considerably associated with C3, rural and urban residence, and income of RON 3001–4000, RON 4001–7000, and > RON 10,000 (r = 0.914–0.995, *p* < 0.05).

Around 82.28% of participants confirmed their availability for eco-food recommendations, while only 1.90% disclaimed it (Figure 6B).

The places of eco-food acquisition frequencies compared to all variable parameters are displayed in Figure 6C,D. A bachelor’s degree highly correlates with an income of RON 2001–3000 and RON 3001–4000/month, and S1, S3, and S5 (r = 0.887–0.989, *p* < 0.05), while a post-college degree is strongly associated with S4 (r = 0.888, *p* < 0.05, Figure 6C and Appendix A). A high correlation exists between ages 25–65 and both sexes, S1, S3–S5, and “yes” for eco-food recommendation (r = 0.892–0.991, *p* < 0.05, Figure 6D and Appendix A). Obesity and normal weight are strongly associated with males, age = 18, S1, and “abstention” for eco-food recommendation (r = 0.909–0.999, *p* < 0.05).

All statistically significant differences between variable parameters correlated with these essential aspects are illustrated in the heatmaps in Appendix A.

#### 3.6.1. Education Level

The relative frequencies (%) of the following parameters increase inversely proportionally to the educational level, in the order academic, college/post-college, and high school/middle school education: perception of eco-food as healthier than conventional ones (90.23% vs. 94.74% vs. 100%, *p* > 0.05), significant/mild eco-food contribution to human health (85.94% vs. 100%, *p* < 0.05), frequently/regularly update rate of eco-food production and provenance (17.58% vs. 54.39% vs. 66.67%, *p* < 0.05), trust in Romanian certified organic foods (32.42% vs. 80.70% vs. 100.00%, *p* < 0.05), constantly verifying the eco-food during acquisition regarding shelf-life (16.41% vs. 54.39% vs. 66.67%, *p* < 0.05) and ingredients (22.66% vs. 78.95% vs. 100.00%, *p* < 0.05), eco-food acquisition monthly/occasionally (39.06% vs. 80.70% vs. 100.00%, *p* < 0.05), eco-food recommendation—yes/abstention (98.05% vs. 98.25% vs. 100%, *p* < 0.05), overweight/obese (53.52% vs. 0.00% *p* < 0.05), and age 25–49 ( 83.59% vs. 52.63% vs. 0.00%, *p* < 0.05).

On the other hand, the relative frequencies of other perceptions decrease directly proportionally to the study level: eco-food quality perception (88.28% vs. 35.09% and 33.33%, *p* < 0.05), daily/weekly eco-food acquisition (55.86% vs. 19.30% vs. 0.00%, *p* < 0.05), and strong confidence rate (C4/C5) in the eco-food label (26.56% vs. 14.04% vs. 0.00%, *p* < 0.05).

#### 3.6.2. Monthly Incomes

The monthly income varies widely (<RON 2000 and >RON 10,000); most participants have RON 4001–7000 (86/316, 27.22%) and >RON 10,000 (77/316, 24.37%). Even if there is no direct proportionality between monthly income and various parameters, the following aspects are significant: the higher price of organic foods is claimed despite the monthly income value (68.09–100.00%). The highest monthly incomes are received by respondents of 25–34 years (RON 7001–>10,000, 48.78–50.65%), while participants in the 35–49 years age group receive the lowest income (<RON 2000–7000, 37.50–60.98%). All monthly income values substantially belong to academic studies (70.83–85.37%). Employees receive RON 2001–>10,000 monthly income (55.52–73.26%), and 54.17% of students have <RON 2000. All incomes are mainly in women from urban zones (51.22–82.98%). The same is available for eco-food higher quality perception (63.83–90.84%), the environmental positive impact of eco-foods (83.11–98.84%), significant eco-food contribution to human health (74.03–100.00%), and eco-foods recommendation (91.49–100.00%). Respondents with RON 2001–3000 purchased eco-foods daily/weekly (43.90–66.67%), while all others (<RON 2000, RON 3001–>10,000) opted for monthly/occasional acquisition (43.90–66.67%). The most substantial eco-food satisfaction level (S4/S5) is available for RON 7001–10,000 and RON 3001–4000 (63.41% and 73.17%). Despite the various monthly incomes and high-price perceptions, the rate of non-consumers of eco-food is very low (13/316, 4.11%), and only 6/316 (1.90%) negatively reacted to the potential organic food recommendation.

#### 3.6.3. Age

Participants of all age groups are mainly employees (54.55–62.73%) from urban zones (61.54–85.75%). A total of 65.45% of 25–34-year-olds and 33.33–37.31% of 35–>50-year-olds are overweight/obese.

The age groups 18–24 and 35–>50 considerably believe in the health benefits of organic food (95.52–100.00%), despite their perception of higher prices (89.74–92.54%). The age group 35–>50 reveals considerable trust (C4/C5) in eco-food labels (33.58–48.72%), while 25–34-year-olds evidence a moderate one (C3, 73.64%). Strong confidence in Romanian-certified organic food and a daily/weekly organic food acquisition are reported for 24–>50-year-olds (48.51–58.18%), while the 18–24 age group opted for monthly/occasional organic food purchases (75.76%). All age groups reported high satisfaction levels with organic food consumption (S4/S5, 51.28–55.97%) and potential organic food recommendations (95.52–100.00%).

#### 3.6.4. BMI Value

Two groups of respondents—underweight/normal weight (179/316, 56.65%) and overweight/obese (137/316, 43.35%)—were differentiated according to BMI values. The incomes are RON 4001–7000 (29.61%) vs. > RON 10,000 (29.20%). Most are women (68.72% > 55.47%, *p* < 0.05) from urban zones (79.33% < 82.48%, *p* > 0.05).

Significant differences were recorded between them regarding whether eco-food is healthier than conventional food (98.32% > 81.75%. *p* < 0.05), the perception of higher quality of eco-food vs. conventional food (69.27% < 89.78%, *p* < 0.05), increased environmental positive impact (96.09% > 86.13%, *p* < 0.05), higher price perception (90.50% < 98.54%, *p* < 0.05), enough/moderate data availability regarding eco-food (34.64% < 80.29%, *p* < 0.05), organic food contribution to human health vs. conventional ones (100.00% > 73.72%, *p* < 0.05), confidence in Romanian-certified eco-food (72.63% > 1.46%, *p* < 0.05), organic food acquisition daily/weekly (16.76% < 90.51%. *p* < 0.05), verification rate during organic food acquisition—shelf-life (occasionally/rarely 46.37% < 98.34%, *p* < 0.05) and ingredients (occasionally/rarely 29.61% < 98.54%, *p* < 0.05), and eco-food recommendation—yes/abstention (97.77% < 98.54%, *p* > 0.05).

#### 3.6.5. Residence

Descriptive analysis reveals several differences between participants from rural (61/316, 19.30%) and urban (255/316, 80.70%) residences: eco-food has a higher quality (70.49% < 80.00%, *p* < 0.05), a positive environmental impact (88.52% vs. 92.55%, *p* > 0.05), and higher prices (91.80% < 94.51%, *p* > 0.05), and eco-food information is not enough (40.98% < 46.67%, *p* > 0.05), there is high confidence in eco-food labels (C4/C5, 19.67% < 25.10%), and eco-food contributes to health (85.25% < 89.41%, *p* > 0.05). On the other hand, the rural respondents believe that eco-food is healthier than conventional food (93.44% > 90.59%, *p* > 0.05), and only one respondent (1/61, 1.64%) avoids eco-food consumption, compared to 12/255 (4.71%) from urban zones. The satisfaction level S4/S5 is higher (59.02% > 53.73%, *p* > 0.05), and the availability (yes/abstention) to recommend eco-food is similar (98.36% vs. 98.04%).

## 4. Discussion

The perceived benefits of organic foods, trust in scientists, communicator credibility, preexisting beliefs, and science-related events (e.g., COVID-19) were significant predictors of the public perception of scientific information about organic foods.

Human living standards have improved significantly in recent decades, and the continuous demand for a better lifestyle and healthier food has also increased. Organic product consumption is an emerging trend, and consumers want to know the benefits of these foods before making purchasing decisions. The present study had the following main objectives:▪Investigate the level of knowledge and familiarity regarding eco-food;▪Understand the respondents’ general attitudes towards organic foods and the factors influencing these attitudes;▪Explore the trust and motivation behind the decision to buy eco-food;▪Analyze the most important factors that determine whether consumers purchase organic food or not;▪Examine the level of satisfaction of the respondents towards the ecological products;▪Correlate these data with sociodemographic data.

### 4.1. Trust in Eco-Foods

Literature data regarding the quality of foods obtained using organic and conventional agriculture confirm the following essential aspects:▪Organic foods have a lower risk of synthetic pesticide contamination;▪Organic foods positively act on the environment and human health;▪No differences were reported regarding heavy metals, mycotoxins, and susceptibility to microbial contamination;▪Comparable safety and nutritional value.

Most participants stated that the current information about organic food was not enough. Their current information sources are supermarkets, advertising, the internet, and family/friends. Numerous studies have analyzed the important role of consumers’ awareness of organic food products in various settings and conditions [84]. Social media is essential to people’s daily lives and can spread awareness of significant information [85]. Therefore, social media influencers may substantially orient consumer behavior to organic food acquisition, increasing its credibility in eco-food value [86]. Moreover, consumers can communicate with professionals in various domains on social media to clarify their concerns about organic foods, gain trust in their benefits, and express personal preferences [87,88].

Trust in organic foods is complex, and various factors influence consumer beliefs. The key aspect of trust in eco-foods is perception regarding environmental concerns and health benefits.

Organic farming practices are considered more sustainable. They reduce the environmental impact through crop rotation, diminish soil erosion, and lower greenhouse gas emissions. Most people believe organic foods are healthier due to their lower pesticide exposure, potentially higher nutrient content, and reduced risk of antibiotic-resistant bacteria.

Most respondents know the beneficial impact of eco-food products on the environment (54.75%) due to organic farming [89] and consider that they are healthier (61.39%) than conventional ones [90]. This last statement shows a strong foundation of trust in organic food due to significantly higher levels of pharmacologically active metabolites, vitamins, and minerals [91,92,93,94,95]. Most respondents perceive organic products to be of higher quality, as numerous studies demonstrated by quantifying the bioactive constituents [96]. The considerably higher price is justified by the rigorous processes involved in organic farming, low yields, and considerable taxes for eco-food certification [97,98,99,100]. In the present study, eco-foods are considered to have an inaccessible price, even by participants with substantial incomes.

The reasons for the skepticism are price, nutritional equivalence, labeling, and certification. Organic foods are more expensive than conventional ones, leading to concerns about affordability and accessibility. More than 65% of respondents cite the high prices of organic food products as the main reason. Some studies have shown minimal differences in nutrient content between organic and conventional products, raising questions about the health benefits. Our respondents are not convinced that organic foods significantly contribute to health compared to conventional ones due to a lack of information about the benefits. This result confirms the previous studies in which the correlation between organic food consumption and health benefits remains insufficiently demonstrated in epidemiological studies [101,102]. While organic standards prohibit synthetic pesticides, some naturally occurring substances used in organic farming can also be harmful. Many concerns exist about the accuracy of organic labeling and the effectiveness of certification processes. A lack of trust in eco-food quality is a considerable obstacle to purchasing organic food products. Furthermore, significant suspicion among consumers regarding the authenticity and incontestable quality of the organic products on the market is a substantial reason. Although the present study evidences a considerable trust in organic food labels (40.25%), almost two-thirds of respondents have no trust or are unsure about organic certifications in Romania.

Almost half of the respondents consider the information available only moderate, and a large percentage consider it insufficient, suggesting that current information efforts are not sufficiently effective or pervasive. Increasing awareness about the benefits and limitations of organic food can empower consumers to make informed choices. This aspect could be improved if people could easily access detailed educational materials such as guides, brochures, interactive websites, informative videos, and collaboration with nutritionists, doctors, and influencers to explain the benefits of organic food in a clear and accessible way.

### 4.2. Motivation for Eco-Food Acquisition and Consumption

Trust in organic foods is multifaceted and influenced by individual values, perceptions, and available information. While organic foods offer potential benefits, it is essential to consider both the positive aspects and possible limitations in motivating their consumption. With economic and social factors, trust in organic foods could influence acquisition, consumption, and preferences.

First, carbohydrates, fats, and other nutrients (vitamins, proteins, minerals, enzymes, energy, etc.) must function optimally. Second, various diseases (obesity, diabetes, heart disease, cancer, osteoporosis, dental diseases, etc.) need a healthier diet. Other factors involve a higher degree of consciousness regarding food’s nutritional and energetic value or the need for spiritual satisfaction after consuming food and dishes, in addition to the basic vital needs.

Price, taste, and flavor are the most common criteria for eco-food acquisition. They are followed in decreasing order by provider and country, product appearance, friend/family recommendations, and eco-friendly packages.

Concern for their health is the main priority for most attendants for acquiring eco-food. With numerous respondents (80.06%), this aspect significantly outperforms other reasons, indicating that consumers strongly emphasize eco-food’s benefits on health. Previously published studies revealed that organic crops have substantial amounts of antioxidant metabolites, reducing the risk of neurodegenerative [91], cardiovascular [103], and other chronic diseases [104,105]. The following essential motivations are the high quality of eco-food and limiting pesticide and additive use in organic food production. This denotes that many consumers perceive eco-food to be of better quality than conventional ones, namely, their concern about reducing exposure to harmful chemicals and artificial additives in their daily diet.

The responses of numerous survey participants suggest a tendency to support local farmers by preferring their eco products and awareness and desire to diminish the negative impact on the environment through sustainable food choices. With 80.06% of respondents, this aspect significantly outperforms other reasons, indicating that consumers strongly emphasize eco-food’s benefits for health.

Numerous studies have analyzed different aspects of organic food packaging (materials, design, size); however, the impact of packaging transparency was less investigated [106]. A recent study explored how transparency in organic food packaging affects consumers’ purchasing intentions and suggested practical solutions for companies [57].

The main reasons to avoid organic food acquisition and consumption are, in decreasing order, high prices, low availability, lack of trust in organic food certification and labeling, and missing data about eco-food benefits for human health.

Mistrust in the control system and doubt about the authenticity of food sold as organic have a substantial negative impact on self-reported buying behavior [60]. Furthermore, numerous studies investigated the effect of organic labels on consumers’ perception of food products [107]. One of the major concerns of consumers is labeling effectiveness [108], which influences the perception of organic food [109]. Moreover, increasing the percentage of consumers with positive attitudes must supplement the available data associating organic labels with the Nutri-Score.

### 4.3. Eco-Food Acquisition Behavior

Our results reveal a significant interest in eco-food among participants, as about 65% of respondents purchase it frequently (daily, weekly, and monthly).

Fruits, vegetables (78.16%), dairy products, and eggs (65.82%) are the most frequently purchased organic food categories (*p* < 0.05). Morna et al. reported similar data [110]. Honey and other healthy foods (53.48%), fish and meat (34.49%), and novel foods (chia seeds, protein powders, microalgae, noni, acai, etc., 25.31%) are also important to consumers (*p* < 0.05). In contrast, basic foods (oil, vinegar, flour, sugar, bread) and sweets are purchased less often in the eco version (14.48% and 7.37%, *p* < 0.05) [27].

The primary sources of purchase of organic food products for most respondents (74.36%) are leading retailers (supermarkets and hypermarkets); however, autochthonous producers cannot access the big retail chains to sell their organic products to customers [44]. The leading supermarkets commonly import vast amounts of food and commercially available organic products [111]. Then, to support local farmers, other participants opt for local markets (49.36%). Pharmacies and health food stores are selected by 26.58% of respondents, neighborhood stores are commonly frequented by 7.91%, 53.77% opted for online acquisitions, and 48% indicated other sources. Another study reports similar preferred places for organic product acquisition [112,113,114,115].

According to the literature reviewed, consumers perceive organic fresh food as more beneficial than conventional ones and are willing to pay additional money. Taste and odor are the most common criteria for eco-food acquisition (N > 200, Figure 4D). For fresh organic foods (fruits and vegetables), higher levels of antioxidants and various nutrients could intensify the taste and flavor [116]. Provider and country (N = 176), aspect (N = 144), and friend/family recommendations (N = 120) are other significant criteria, while 40 respondents mention eco-friendly packages (*p* < 0.05). Organic food packaging is essential in preserving product quality, supporting sustainability, and building consumer trust in the producer company. Clear labeling (certifications and sustainability information) builds trust in the product’s authenticity. It often uses materials that minimize environmental impact, such as biodegradable, compostable, or recyclable materials. Eco-friendly packaging is paper-based, bioplastic, and glass; they must balance sustainability with functionality (moisture resistance and durability). Consumer expectations from these eco-friendly packages are transparency, alignment with organic values, and aesthetic consideration for a high-price food category [117].

### 4.4. Limitations

Even though online surveys are becoming increasingly popular because they are convenient, easy, and inexpensive data collection tools [75], our study, based on the Google Form questionnaire, has several limitations. The accuracy of the responses cannot be verified because the study database consists of self-reported information on organic foods. This study was conducted for three months; then, due to the condition of voluntary participation, the sample size cannot be predicted. The 316 respondents to all 30 questions did not represent the Romanian population due to the probabilistic selection of individuals who wanted to fill in an online form.

### 4.5. Essential Considerations

Our results reveal that the organic food concept is still not correctly perceived due to the lack of suitable information and transparency. Most consumers are moderately satisfied with organic food consumption (S3), but a significant segment remains neutral, offering opportunities to increase satisfaction through improving the current organic products and sharing their benefits.

People with high educational levels (academic and post-college) report significant satisfaction with organic food consumption (S4 and S5). There is also a high correlation between ages 25–65, moderate–high satisfaction (S3–S5), and “yes” for eco-food recommendations. Moderate to high satisfaction levels (S3–S5) are also associated with moderate confidence in eco-food labels (C3) and moderate to high income. Our results show that monthly income and residence are not essential factors in higher price perception. Insignificant price variation perception correlated with C4 and weekly acquisition. Similar price perception substantially correlates with C5 and daily acquisition. Lower price perception strongly correlates with minimal confidence and monthly acquisition. Our study results are similar to those from previously published ones and verify the hypotheses. Literature data show that gender, age, and education differentiate the criteria influencing the purchasing of organic foods [118]. Young (18–25 years) and middle-aged women (35–60 years) with academic education highlight the importance of food safety, quality, and nutritional value [119]. Significant differences between the sexes appear in the field of bioactive compounds, which are more important for women than for men. Young consumers are familiar with high technology and are more receptive to its food processing applications. Motivations are differentiated by age and gender; women and older men are generally more interested in organic food safety and care more about their health [95]. Literature data also confirm that individuals with a lower BMI (underweight and normal-weight) have healthier diets and trust more in the organic food benefits [119]. Monthly income cannot explain the differences in organic food purchasing behavior [120].

“Organic” does not always mean “healthy” because processed eco-foods are still high in sugar, fat, and calories and have a lower shelf-life [121]. They may expose consumers to the risk of nutritional imbalances, sometimes serious, because of the high content of negative-impact compounds (salt, added sugar, and saturated fats). If sugar, salt, and saturated fats, although organic certified, are added by manufacturers in exceedingly large quantities, with the primary goal of obtaining products that entice many senses and create consumer dependency, the risk of nutritional imbalance is as high as for conventional foods [17]. That is the reason for the verification of organic food labels. Our results also report that <35% of respondents always verify the eco-food ingredients, and <25% check the shelf-life by reading the label from the package. It is significant to show that verifying the organic food’s shelf-life and ingredients during the acquisition process is predominant when respondents perceive eco-food as higher-quality (76.45% and 76.53%), pricier (93.52% and 93.54%), and healthier than conventional ones (90.44% and 90,48%), with benefits for the environment (91.13% and 91.16%) and human health (87.71% and 87.76%); when they have an academic education (83.96 and 79.59%); and when they recommend organic food (97.95% and 98.10%). The same is true for females and urban respondents with high satisfaction levels (54.75–80.89%).

Our findings are similar to those from previous research [16,122]; the Romanian organic product consumers in Romania are generally younger (under 40 years), educated, and urban residents [113,123,124,125]. Women and high-income households are particularly inclined toward organic products [126,127,128]. Higher education levels and environmental awareness correlate with increased organic consumption, associated with superior quality, freshness, and nutritional value [16,129,130]. The primary drivers are health benefits, taste, the absence of harmful chemicals, and environmental sustainability [131,132,133]. High costs remain the most significant barrier to broader adoption [134]. Skepticism toward organic certifications and labeling undermines consumer confidence. Limited availability and inadequate promotion reduce market penetration, especially in rural areas. The shorter shelf lives of organic products diminish consumer interest. Supermarkets are the most common purchase points, but farmer’s markets and specialty stores also play essential roles. There is a preference for local organic products, reflecting support for domestic producers and trust in local certifications [135,136]. Fresh fruits and vegetables are the most purchased, followed by dairy, eggs, and meat [76,137]. Processed organic foods such as cereals, sweets, and canned goods are less popular, partly due to cost and cultural preferences.

It is essential to notice that less self-care consciousness (overweight/obese—46.08% and 46.60%) and substantial confidence in the eco-food label (C4/C5, 25.94%, and 24.05%) and Romanian-certified organic foods (41.77% and 37.20%) decrease the organic food verification during acquisition. Moreover, trust in an organic diet, eco-food certification, and source credibility lead to a substantial appreciation of all organic food benefits. This halo effect of organic food can bias sensory and healthiness perceptions and influence eco-food acquisition behavior [135,138,139,140,141].

The choice of organic foods is correlated to perceived benefits for human health and environmentally friendly behavior; therefore, ecological value is considered a health condition [142,143]. The COVID-19 pandemic focused people’s attention on healthy food, and fresh vegetables or dairy products were purchased directly from local producers [88,137,144,145,146,147,148].

Moreover, organic food consumers could share their concerns, actions, ideas, and experiences to achieve their desires or needs through social media [149]. The eco-food purchase decision is determined by needs, knowledge, perceptions, motivations, values, beliefs, attitudes, and financial resources; digital tools can play an essential role in real-time communication between consumers and local suppliers and in promoting organic food. In the present study, most respondents are young or middle-aged, with academic studies and active workers. Health is the main priority in achieving their professional targets. Most believe organic foods are healthier than conventional ones (91.14%) and that eco-food benefits human health (88.61%). The main reasons are organic food safety, higher quality and nutritional value (85.75%), and limited use of pesticides and additives (79.43%). Personal trust in the positive health effects of organic food consumption leads to regular information updates and rigorous verifying of the eco-food ingredients and shelf-life by reading the labels during acquisition. Furthermore, they reported high satisfaction (S4–S5) with organic food consumption and recommended it. These data suggest the Health Belief Model (HBM) underlines their eco-food purchase behavior.

Organic food communities share common interests, values, and behaviors related to organic food consumption and emphasize sustainability, health, and environmental consciousness. The Theory of Planned Behavior (TPB) can also be applied in this case because organic food acquisition is based on firmly believing that it is appropriate and on the approval of the social circle.

Our respondents claim that environmental concerns underline sustainable food acquisition. They are convinced that organic foods positively impact the environment (91.77%) and that their production is safe and sustainable. They purchase eco-foods from the neighboring market (25/316); organic food acquisition is motivated by support for local farmers, environmental concerns, limited use of pesticides, and eco-friendly packaging (94.62%). In this case, the Value–Belief Norm (VBN) applies because organic foods are considered eco-friendly, positively impacting the environment and preserving biodiversity.

Thus, our study results reveal that all three behavioral models could underline organic food acquisition behavior, also reported by previous findings [59,72,73,150,151,152,153].

## 5. Conclusions and Further Directions

The present study shows that sociodemographic factors differentiate consumers and influence perceptions and motivations for organic food acquisition. The extensive information collected, and a deep analysis of the knowledge, perception, attitude, trust, and motivation involved in Romanian consumer behavior regarding eco-food acquisition could enrich the scientific database.

Overall, organic food products are well regarded by consumers and are recommended with confidence. However, a segment of the population does not feel sufficiently informed or convinced to make a clear recommendation, representing an opportunity for manufacturers and distributors to improve public information and education. The organic food market in Romania is constantly developing, and trends indicate an increase in demand as more and more consumers become aware of the benefits of these products. It is expected that in the future, higher accessibility and the diversification of supply will lead to broader market penetration.

Our findings suggest that policymakers’ involvement in public information, educational campaigns, financial investments, and marketing strategies to support local organic food producers is essential for increasing interest in eco-foods. Furthermore, extensive studies must be conducted on people with various chronic diseases to evaluate the health benefits of organic food and investigate the possibility of affording their daily consumption.

## Figures and Tables

**Figure 1 foods-14-00293-f001:**
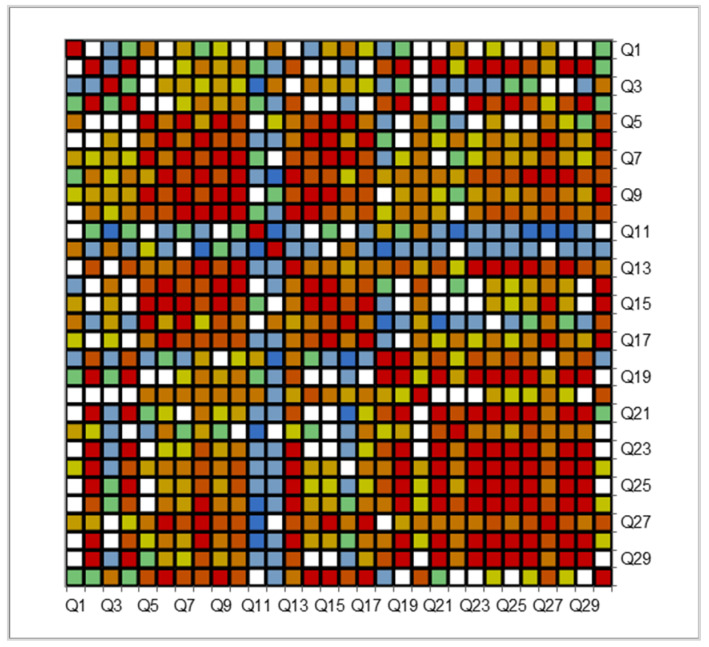
Reliability analysis of the questionnaire—the correlation map of all 30 multiple-choice queries (Q1–Q30) with 316 respondents.

**Figure 2 foods-14-00293-f002:**
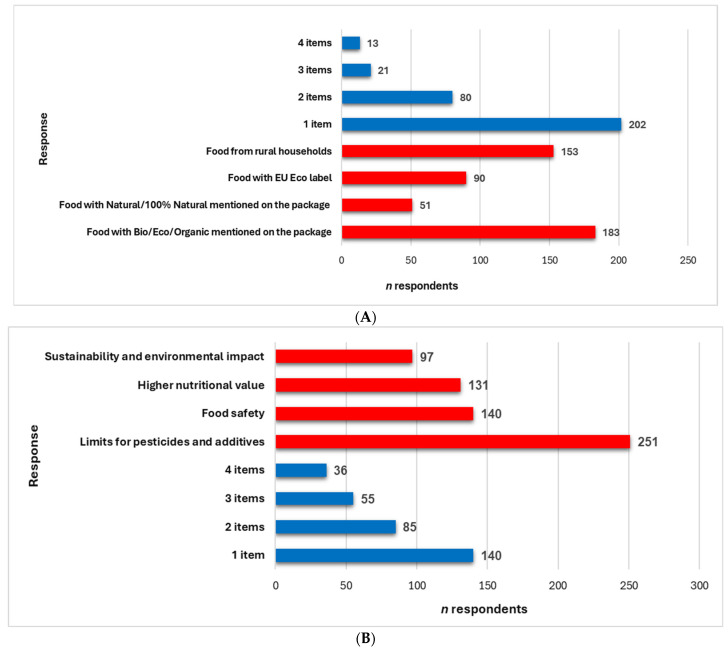
(**A**) Eco-food differentiation. (**B**) Essential aspects linked with eco-food production. The responses are illustrated in red, and the items’ numbers are in blue.

**Figure 3 foods-14-00293-f003:**
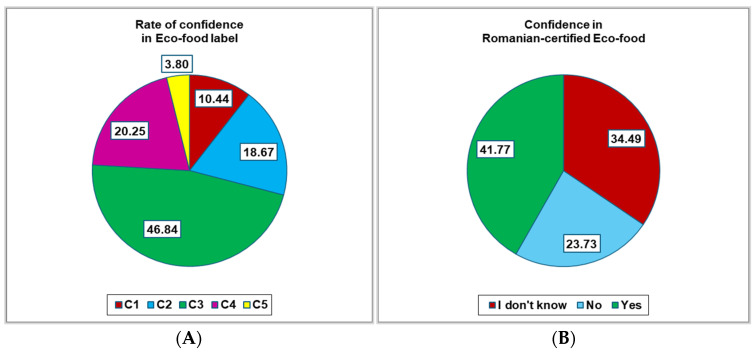
(**A**) Trust in eco-food labels; (**B**) trust in Romanian-certified eco-food; C = confidence; C1–C5, C1 = minimal level of confidence, C5 = maximal level of confidence.

**Figure 4 foods-14-00293-f004:**
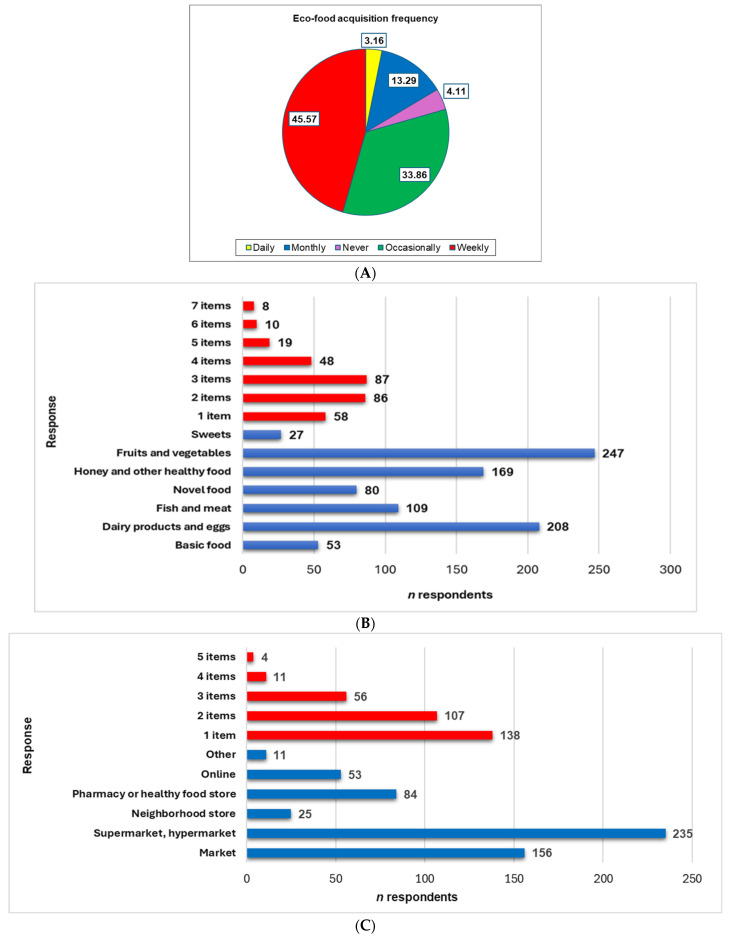
(**A**) The eco-food acquisition frequency; (**B**) the eco-food type preferences; (**C**) the source of eco-food acquisition; (**D**) the main criteria involved in eco-food acquisition. The responses are illustrated in blue, and the items’ numbers are in red.

**Figure 5 foods-14-00293-f005:**
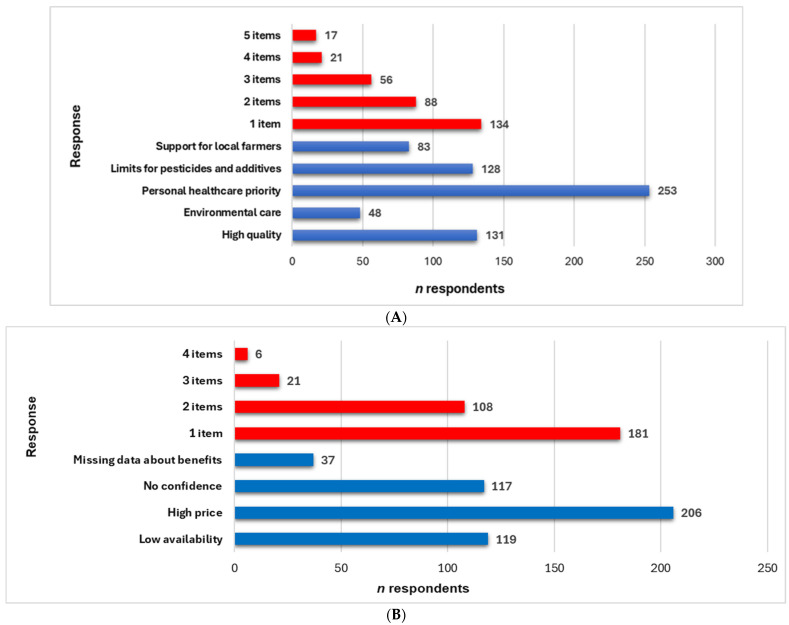
(**A**) The main reasons for eco-food acquisition; (**B**) The main reasons to avoid eco-food. The responses are illustrated in blue, and the items’ numbers are in red.

**Figure 6 foods-14-00293-f006:**
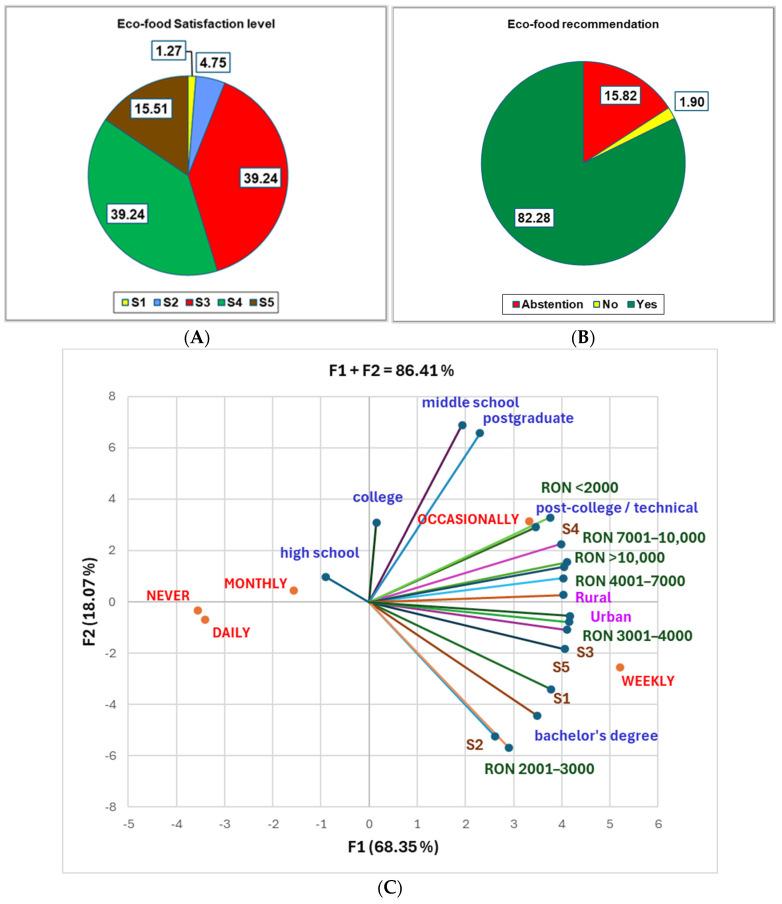
(**A**) Organic food satisfaction level (S1—minimal, S5—maximal); (**B**) organic food potential recommendation; (**C**) the correlations between eco-food acquisition frequency and monthly income, education level, and satisfaction level. (**D**) the correlations between eco-food acquisition frequency and BMI, age group, sex, satisfaction level, and potential recommendation.

**Table 1 foods-14-00293-t001:** Essential differences between organic food consumption between the Western and Northern EU countries and Romania [21,22].

Aspect	Western and Northern European Countries	Romania
Market Size	Mature and extensive	Emerging and niche
Distribution	Well-developed retail networks	Limited, especially in rural areas
Policy Support	Robust and comprehensive	Developing
Awareness	High	Growing but limited
Motivation	Health, sustainability, ethics	Primarily health-driven
Barriers	Price and occasional skepticism	Price, availability, trust

**Table 2 foods-14-00293-t002:** Sociodemographic data of all 316 respondents.

Parameter		Total		F		M		*p*-Value
		*n*	%	*n*	%	*n*	%	
Sex		316.00	100.00	199.00	62.97	117.00	37.03	<0.05
Residence	Rural	61.00	19.39	41.00	20.60	20.00	17.09	
	Urban	255.00	80.70	158.00	79.40	97.00	82.91	
Age	age 19–24	32.00	10.13	24.00	12.06	8.00	6.84	<0.05
	age 25–34	110.00	34.81	77.00	38.69	33.00	28.21	
	age 35–49	134.00	42.41	79.00	39.70	55.00	47.01	
	age 50–65	32.00	10.13	13.00	6.53	19.00	16.24	
	age = 18	1.00	0.32	1.00	0.50	0.00	0.00	
	age > 65	7.00	2.22	5.00	2.51	2.00	1.71	
Study level	Bachelor’s degree	152.00	48.10	90.00	45.23	62.00	52.99	<0.05
	college	40.00	12.66	26.00	13.07	14.00	11.97	
	high school	2.00	0.63	2.00	1.01	0.00	0.00	
	middle school	1.00	0.32	1.00	0.50	0.00	0.00	
	post-college/technical	17.00	5.38	12.00	6.03	5.00	4.27	
	postgraduate	104.00	32.91	68.00	34.17	36.00	30.77	
Occupation	employee	195.00	61.71	123.00	61.81	72.00	61.54	<0.05
	entrepreneur/owner	47.00	14.87	31.00	15.58	16.00	13.68	
	homeworker	26.00	8.23	14.00	7.04	12.00	10.26	
	pensioner	8.00	2.53	6.00	3.02	2.00	1.71	
	self-employed	16.00	5.06	10.00	5.03	6.00	5.13	
	student	20.00	6.33	13.00	6.53	7.00	5.98	
	unemployed	4.00	1.27	2.00	1.01	2.00	1.71	
Income	RON 2001–3000	47.00	14.87	27.00	13.57	20.00	17.09	<0.05
	RON 3001–4000	41.00	12.97	21.00	10.55	20.00	17.09	
	RON 4001–7000	86.00	27.22	61.00	30.65	25.00	21.37	
	RON 7001–10,000	41.00	12.97	27.00	13.57	14.00	11.97	
	<RON 2000	24.00	7.59	15.00	7.54	9.00	7.69	
	>RON 10,000	77.00	24.37	48.00	24.12	29.00	24.79	
BMI	Normal weight	134.00	42.41	92.00	46.23	42.00	35.90	<0.05
	Obese	45.00	14.24	16.00	8.04	29.00	24.79	
	Overweight	92.00	29.11	60.00	30.15	32.00	27.35	
	Underweight	45.00	14.24	31.00	15.58	14.00	11.97	

F—female, M—male, *p*-value < 0.05 indicates significant statistical differences, BMI—body mass index value and its significance expressed as normal weight, obese, underweight, and overweight.

## Data Availability

The original contributions presented in the study are included in the article/Appendix A, further inquiries can be directed to the corresponding authors.

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
