# Peer review of "Perception, Trust, and Motivation in Consumer Behavior for Organic Food Acquisition: An Exploratory Study"

_foods, 2025, doi:10.3390/foods14020293_

Round 1

Reviewer 1 Report (Previous Reviewer 2)

Comments and Suggestions for Authors

Thank you for the opportunity to review the revised version of your manuscript.

Please find bellow my comments and suggestions to improve it.

1) Abstract

The abstract has been improved.

2) Introduction section

This section is too focused on EU regulations. However, it should present the research gaps based on the outcomes of previous studies.

It should describe the findings of related studies that motivate carrying out this study.

3) Literature review

The manuscript lacks a literature review in which the related studies' findings should be described. The study analyzes antecedents of food consumption behavior like trust, perception, etc. The literature review section should present what previous research has investigated about these antecedents regarding organic food consumption.

4) Absence of theory

Is there a behavior theory that grounds this study? There are several behavior theories that could have been adopted in this study to explain how the research model was defined.

5) Methodology

The manuscript fails to describe the related studies that inspired the survey questionnaire elaboration. It should present the references used to define the questionnaire questions.

6) Results

Some changes have been made to better and more thoroughly present the findings. However, the link between the antecedents (perception, trust, and motivation) could be clearer. For example, in the Results section there is just one mention to the word "trust" in a question (Figure 2A) in which it is not clear the relationship between trust and that question. I still think the lack of a literature review to clearly present these concepts is a major drawback of the current version of this study.

Author Response

Comments and Suggestions for Authors

Thank you for the opportunity to review the revised version of your manuscript.

Please find bellow my comments and suggestions to improve it.

Q1) Abstract

The abstract has been improved.

R1. Thank you so much for appreciating our revised abstract.

Q2) Introduction section

This section is too focused on EU regulations. However, it should present the research gaps based on the outcomes of previous studies.

It should describe the findings of related studies that motivate carrying out this study.

Q3) Literature review

The manuscript lacks a literature review in which the related studies' findings should be described. The study analyzes antecedents of food consumption behavior like trust, perception, etc. The literature review section should present what previous research has investigated about these antecedents regarding organic food consumption.

Q4) Absence of theory

Is there a behavior theory that grounds this study? There are several behavior theories that could have been adopted in this study to explain how the research model was defined.

R2-R4 Thank you so much for these attentive and excellent comments. The revised Introduction section contains 4 subsections:

  • Literature review (lines 70-170)
  • Hypotheses (lines 174-184)
  • Behavioral theories (lines 186-201
  • Aim of the present study (lines 204-211)

Q5) Methodology

The manuscript fails to describe the related studies that inspired the survey questionnaire elaboration. It should present the references used to define the questionnaire questions.

R5. Thank you so much for this insightful comment. The requested data are in lines 216-218, with 4 references.

Q6) Results

Some changes have been made to better and more thoroughly present the findings. However, the link between the antecedents (perception, trust, and motivation) could be clearer. For example, in the Results section there is just one mention to the word "trust" in a question (Figure 2A) in which it is not clear the relationship between trust and that question. I still think the lack of a literature review to clearly present these concepts is a major drawback of the current version of this study.

R6. The results, discussion and conclusions are extensively reorganized and all changes are marked with yellow (lines 283-793).

  • The last subsection of the Discussion (lines 718-770) presents the essential data of the present study according to hypotheses and theories.
  • Conclusions were revised accordingly.
  • Over 20 references were added

Reviewer 2 Report (Previous Reviewer 3)

Comments and Suggestions for Authors

Thanks for the revisions. In order to provide readers with a better experience, appropriate images are essential. Authors can supply the website of organic food images in the references.

Author Response

Comments and Suggestions for Authors

Q1. Thanks for the revisions. In order to provide readers with a better experience, appropriate images are essential. Authors can supply the website of organic food images in the references.

R1. Thank you for the valuable comment. The images were introduced as requested in the Supplementary material, Figure S1.

Reviewer 3 Report (Previous Reviewer 1)

Comments and Suggestions for Authors

This version of the manuscript is really improved. I have no further comments.

Author Response

Comments and Suggestions for Authors

Q1. This version of the manuscript is really improved. I have no further comments.

R1. Thank you so much for the positive feed-back.

Round 2

Reviewer 1 Report (Previous Reviewer 2)

Comments and Suggestions for Authors

Thank you for the opportunity to review the revised version of your manuscript.

Please find bellow some comments and minor suggestions to improve it before publication.

1) Introduction section

Section 1.1 now establishes a good background to present the motivation to carry out this study. Also, Section 1.2 presents the gaps/hypotheses this study aims to fill.

I would only say that in Section 1.1, there are some paragraphs without the corresponding references. Data in Table 1 should be presented with the corresponding citation(s).

2) Literature review

It is now presented in Section 1. It has a focus on Romanian organic food market, but at least, it is not connected to the gaps/questions this study deals with.

3) Absence of theory

Three consumer behavior theories are now mentioned in the Introduction section. However, it is still necessary to articulate them with the background and results of this study. In the Discussions section, please discuss how the findings of this study contribute to the development of the chosen theory(ies).

4) Methodology

The manuscript now cites the related studies that inspired the survey questionnaire elaboration.

5) Results

Some changes have been made to better and more thoroughly present the findings. However, the link between the antecedents (perception, trust, and motivation) could be clearer by defining these concepts, maybe in Section 2.1.

6) Discussion

This section has enriched the presentation of the study's findings. Congratulations on all your effort!

Author Response

1) Introduction section

Section 1.1 now establishes a good background for explaining the motivation for conducting this study. Section 1.2 also presents the gaps/hypotheses this study aims to fill.

Q1 I would only say that in Section 1.1, there are some paragraphs without the corresponding references. Data in Table 1 should be presented with the corresponding citation(s).

R1. All requested references were added and the paragraphs ghanged were marked with yellow. Our MS now has 154 references

2) Literature review

Q2 It is now presented in Section 1. It focuses on the Romanian organic food market, but at least it is not connected to the gaps/questions this study addresses.

R2 We restructured data from Table 1 and the Literature review and separated it into 2 subsections for better understanding. (lines 82-152)

In Materials and Methods, the connection with the gaps is made in lines 212-225

3) Absence of theory

Q3. Three consumer behavior theories are now mentioned in the Introduction section.

However, it is still necessary to articulate them with the background and results of this study. In the Discussions section, please discuss how the findings of this study contribute to the development of the chosen theory (ies).

R3. Done:

In the background: lines 134-152

In the results: lines 291-298, 315-322, 375-382, 386-390, 448-462, 491-517

All aspects are discussed in lines 552-585, 631-650, 664-665, 681-695, 717-725, 735-738, 740-812.

4) Methodology

R4. The manuscript now cites the related studies that inspired the survey questionnaire elaboration.

Q4. Thank you for the positive feed-back.

5) Results

R5. Some changes have been made to better and more thoroughly present the findings. However, the link between the antecedents (perception, trust, and motivation) could be clearer by defining these concepts, maybe in Section 2.1.

Q5. The link is presented in Lines 212-224 from Section 2.1.

6) Discussion

R6. This section has enriched the presentation of the study's findings. Congratulations on all your effort!

Q6. Thank you so much for everything!

This manuscript is a resubmission of an earlier submission. The following is a list of the peer review reports and author responses from that submission.

Round 1

Reviewer 1 Report

Comments and Suggestions for Authors

In my point of view, after some revisions, the work submitted by Morosan and collaborators can be considered for publication in Foods after the following adjustments:

The word limit for the abstract is 200 words and it should be structured in Background; Methods; Results; and Conclusion. In the results, you have to highlight the obtained values with statistical significance.

Lines 140-141: This information about the questionnaire shouldn’t be given here, please move it to Materials and Methods.

In the Materials and Methods section, clarify how you validated your questionnaire, did you pre-test it? Can you provide it as a supplementary material? This section needs to be improved and more details should be given. It is very poor in its current state. How did you calculate your sample size to find it representative?

The Reliability analysis you mention in section 3 is not specified in the Materials and Methods. This has to be revised.

You present a lot of images but further explanations should be given regarding them, like the example of Figure 2.

The Discussion section is fine but I miss a Conclusions section to highlight the relevance of your study. What can the readers learn from your research? What are the practical implications for the scientific community, policymakers, and populations from an international perspective?

Author Response

Q1. The word limit for the abstract is 200 words and it should be structured in Background; Methods; Results; and Conclusion. In the results, you have to highlight the obtained values with statistical significance.

R1. The authors are grateful for the Reviewer 1 comment. All requested revisions are performed in lines 31-47, as follows:

Background. A sustainable healthy diet assesses human well-being in all life stages, protects envi-ronmental resources, and preserves biodiversity. This work investigates the sociodemographic factors, knowledge, trust, and motivations involved in organic food acquisition behavior. Methods: An online survey via the Google Forms platform, with 316 respondents, was conducted from 01 March to 31 May 2024. Results: Our findings show that suitably informed people with high edu-cational levels (academic and post-college) report significant satisfaction with organic food con-sumption  (p<0.05). There is also a considerable correlation between ages 25-65, moderate to high satisfaction, and "yes" for eco-food recommendations (p<0.05). The same satisfaction levels are also associated with medium confidence in eco-food labels and moderate to high monthly income (p<0.05). Our results show that monthly income and residence are not essential factors in higher price perception. Insignificant price variation perception correlated with high confidence and weekly acquisition (p<0.05). Similar price perception correlates with the highest confidence level and daily acquisition (p<0.05). Obese respondents exhibited minimal satisfaction and opted for "abstention" from eco-food recommendations (p<0.05). Conclusions: The present study offers a deep analysis of Romanian people's knowledge, perception, and confidence regarding organic foods. It demonstrates that sociodemographic factors differentiate consumers and influence attitudes and motivation for organic food acquisition.

Q2. Lines 140-141: This information about the questionnaire shouldn’t be given here, please move it to Materials and Methods.

R2. Done, as follows:

Lines 132-136, the aim of the present study: Given the growing interest among the population in organic products, we considered it necessary to conduct an online survey aiming to explore the perceptions and preferences of Romanian people. The data collected could enrich the current scientific database by deeply analyzing the knowledge, perception, attitude, trust, and motivation involved in Romanian consumer behavior regarding eco-food acquisition.

Lines 138-157, Methods:

2.1. Online Questionnaire Presentation

The survey involved voluntary participants ≥ 18 years old residing in Romania. It was approved by the Ethics Committee of the Faculty of Pharmacy, Carol Davila University of Medicine and Pharmacy (Document No. 14357, approved on 30 May 2024). The questionnaire, including 30 multiple choice queries, was distributed through online platforms between 01 March and 31 May 2024, and data was collected electronically in a Microsoft 365 Excel v. 2024 workbook. Thirty questions were generated in electronic format on the Google Form platform. The research team members distributed the URL link via email, SMS, or social and professional networks to colleagues, relatives, and personal contacts. Participants were informed about the survey's aim, the research team involved, and the time required to complete the questionnaire; moreover, they were assured that any email address was collected and that the General Data Protection Regulation (GDPR) guarantees the confidentiality of sensitive personal information. Then, they completed and signed the participation agreement and the individual consent form to enable the publication of research results. During three months, 316 Romanian residents responded to all 30 questions.

The questionnaire was structured in three distinct parts. The first questions aim to collect the participants' sociodemographic data. The second part analyzes their perception and understanding of the eco-food concept. The third part investigates the behavior of acquisition and consumption of eco-food products.

Q3. In the Materials and Methods section, clarify how you validated your questionnaire, did you pre-test it? Can you provide it as a supplementary material? This section needs to be improved and more details should be given. It is very poor in its current state. How did you calculate your sample size to find it representative?

R3. The authors thank for the Reviewer 1 constructive comments.

  • The study is based on an online questionnaire conducted in 3 months, with voluntary participation. Thus, the authors could not establish the sample size, due to this reason. Moreover, beying an online survey, based on self-reported data, the accuracy of the responses is not ensured. The authors evaluated the reliability of the questionnaire, which is an essential condition of its validity, evidencing the substantial quality of the survey.

  • The authors’ responses are detailed in “Limitations” supported by suitable reference, as follows:

Lines 558-565:

Even though online surveys are becoming increasingly popular because they are convenient, easy, and inexpensive data collection tools [76], our study, based on the Google Form questionnaire, has several limitations. The accuracy of the responses can not be verified because the study database consists of self-reported information on organic foods. The study was conducted for three months; then, due to the condition of voluntary participation, the sample size can not be predicted. The 316 respondents to all 30 questions did not represent the Romanian population due to the probabilistic selection of individuals who wanted to fill in an online form.

Significant data are included in the Supplementary Material, which contains the questionnaire with all 30 querries, grouped in 3 categories, and Reliability Analysis extensively described in the excell document.

Q4. The Reliability analysis you mention in section 3 is not specified in the Materials and Methods. This has to be revised.

R4. The authors appreciate this valuable observation and revised as requested.

Lines 159-163:

2.2. Reliability Analysis

The questionnaire was investigated using the Reliability Analysis Internal Model from XLSTAT Life Sciences v. 2024.3.0.1423 by Lumivero (Denver, CO, USA) [29]. Cronbach's alpha index and Guttman L1-L6 coefficients were calculated.

Q5. You present a lot of images but further explanations should be given regarding them, like the example of Figure 2.

R5. All Figures represent original graphs obtained using various tools of Statistical Analysis. The authors verified the MS text and all explanations were included in the rejected Manuscript ID foods-3378425. With the Reviewer 1 permission, they would like to indicate the line number for all explanations:

  • Lines 208-225 for Figure 2;
  • Lines 247-268 and 275-299 for Figure 3;
  • Lines 307-319 and 328-333 for Figure 4;
  • Lines 335-351 for Figure 5.

The authors also added supplementary explanations for Figure 1 (lines 173-177).

Q6. The Discussion section is fine but I miss a Conclusions section to highlight the relevance of your study. What can the readers learn from your research? What are the practical implications for the scientific community, policymakers, and populations from an international perspective?

R6. The authors appreciate the valuable Reviewer 1 comments.

With the Reviewer 1 permission, they want to evidence that the current EU and Romanian regulations regarding organic farming and eco-foods certification are extensively detailed in the Introduction section (Lines 61-131)

Various practical implications are already evidenced in the Discussion section (lines 388-390, 405-413, 419-421, 446-459, 503-526).

They reorganized the Discussion section, separated and completed the Conclusions as follows (lines 572-590):

The present study shows that sociodemographic factors differentiate consumers and influence perceptions and motivations for organic food acquisition. People with high educational levels (academic and post-college) report significant satisfaction with organic food consumption (S4 and S5). There is also a high correlation between ages 25-65, moderate-high satisfaction (S3-S5), and "yes" for eco-food recommendations. Moderate to high satisfaction levels (S3-S5) are also associated with moderate confi-dence in eco-food labels (C3) and moderate to high income. Our results show that monthly income and residence are not essential factors in higher price perception. In-significant price variation perception correlated with C4 and weekly acquisition. Similar price perception substantially correlates with C5 and daily acquisition. Lower price perception strongly correlates with minimal confidence and monthly acquisition. The present study could enrich the current scientific database due to the extensive information collected and a deep analysis of knowledge, perception, attitude, trust, and motivation involved in Romanian consumer behavior for eco-food acquisition. Our findings suggest that policymakers involvement in ensuring public information, educational campaigns, financial investments, and marketing strategies to support local organic food producers is essential for increasing interest in eco-foods. Furthermore, extensive studies must be conducted on people with various chronic diseases to evaluate the health benefits of organic food and investigate the possibility of affording their daily consumption.

Reviewer 2 Report

Comments and Suggestions for Authors

General

- Several figures are referenced by a number and a letter (example 6B), but the figure caption is just B.

Author Response

Q1. General

R1. The authors are grateful to Reviewer 2 for providing such a valuable comment  aiming to increase the quality of the present MS. They would like to mention that all changes are marked with track changes in the revised version.

Q2. Several figures are referenced by a number and a letter (example 6B), but the figure caption is just B.

R2. With the Reviewer 2 permission, the authors would like to mention that Figure 6 (actually, Figure 5) is complex, and formed by 6 statistical graphs ton support the present results, noted from A to F, according to MDPI Instructions. The figure caption explains them, as follows: Figure 5. A, C, E. The frequency of eco-food acquisition (A); The level of personal satisfaction induced by eco-food consumption (C); The potential of eco-food recommendations to other potential consumers (E). B, D, F. Statistically significant differences between variable parameters correlated with all essential aspects (Lines 358-361). All notations belong to the same Figure 5 (Former Figure 6).

Reviewer 3 Report

Comments and Suggestions for Authors

The authors conducted a questionnaire on the demands for organic food. Based on the survey results, this article analyzes the factors that affect people's daily food consumption. The article seems to have not been well organized. Some revisions are needed as follows.

1. The length of the abstract is too long. What do the abbreviations appearing in the abstract mean? The abstract does not need to introduce too much research background, but mainly introduces the research content and conclusions.

2. In the part of Introduction, the growth mode, sources, nutritional value, and correlation with human health of organic food should be introduced in detail. Please provide pictures of typical organic foods.

3. The size of the images is inconsistent and the information is not comprehensive. Please adjust carefully.

4. The article contains too much contents, please summarize them highly. Not all results need to be presented in a pie chart, which feels very cumbersome.

5. The structure of the article is incomplete. What is the final research conclusion?

Author Response

Q1. The length of the abstract is too long. What do the abbreviations appearing in the abstract mean? The abstract does not need to introduce too much research background, but mainly introduces the research content and conclusions.

R1. The authors thank to the Reviewer 3 for significantly valuable comments. They revised the abstract accordingly, as follows in lines 31-47.

Background. A sustainable healthy diet assesses human well-being in all life stages, protects envi-ronmental resources, and preserves biodiversity. This work investigates the sociodemographic factors, knowledge, trust, and motivations involved in organic food acquisition behavior. Methods: An online survey via the Google Forms platform, with 316 respondents, was conducted from 01 March to 31 May 2024. Results: Our findings show that suitably informed people with high edu-cational levels (academic and post-college) report significant satisfaction with organic food con-sumption  (p<0.05). There is also a considerable correlation between ages 25-65, moderate to high satisfaction, and "yes" for eco-food recommendations (p<0.05). The same satisfaction levels are also associated with medium confidence in eco-food labels and moderate to high monthly income (p<0.05). Our results show that monthly income and residence are not essential factors in higher price perception. Insignificant price variation perception correlated with high confidence and weekly acquisition (p<0.05). Similar price perception correlates with the highest confidence level and daily acquisition (p<0.05). Obese respondents exhibited minimal satisfaction and opted for "abstention" from eco-food recommendations (p<0.05). Conclusions: The present study offers an extensive analysis of Romanian people's knowledge, perception, and confidence regarding organic foods. It demonstrates that sociodemographic factors differentiate consumers and influence atti-tudes and motivation for organic food acquisition.

Q2. In the part of Introduction, the growth mode, sources, nutritional value, and correlation with human health of organic food should be introduced in detail

R2. The authors are grateful for these valuable comments. With the Reviewer 2 permission, they would want to mention that in the rejected MS ID foods-3378425 these aspects were already extensively described, following the study design, in lines 63-93, 101-121, 395-403, 432-438, 448-459, 476-490, 494-502, 557-558.

Q3. Please provide pictures of typical organic foods.

R3. The authors did not agree with Reviewer 3 comment, for three main reasons:

  • The study has not external fundings, opffered by an organic food company, to promote their eco-products.
  • The authors aim to analyze the organic food aquisition behavior; they are not eco-food producers, to have original images of organic foods for including in the MS text.
  • The study is conceived as an article, not as a literature review, to include the auxiliar figures with copyright permission.

Q4. The size of the images is inconsistent and the information is not comprehensive. Please adjust carefully. Not all results need to be presented in a pie chart, which feels very cumbersome.

R4. The authors removed the corresponding images.

Q5. The article contains too much contents, please summarize them highly.

R5. The authors attentively revised the MS content. All changes are marked with Track changes.

Q6. The structure of the article is incomplete. What is the final research conclusion?

R6. The authors are grateful for this valuable comment and revised the Conclusions as follows in lines 572-590:

The present study shows that sociodemographic factors differentiate consumers and influence perceptions and motivations for organic food acquisition. People with high educational levels (academic and post-college) report significant satisfaction with organic food consumption (S4 and S5). There is also a high correlation between ages 25-65, moderate-high satisfaction (S3-S5), and "yes" for eco-food recommendations. Moderate to high satisfaction levels (S3-S5) are also associated with moderate confi-dence in eco-food labels (C3) and moderate to high income. Our results show that monthly income and residence are not essential factors in higher price perception. In-significant price variation perception correlated with C4 and weekly acquisition. Sim-ilar price perception substantially correlates with C5 and daily acquisition. Lower price perception strongly correlates with minimal confidence and monthly acquisition.

The present study could enrich the current scientific database due to the extensive information collected and a deep analysis of knowledge, perception, attitude, trust, and motivation involved in Romanian consumer behavior for eco-food acquisition. Our findings suggest that policymakers' involvement in ensuring public information, educational campaigns, financial investments, and marketing strategies to support local organic food producers is essential for increasing interest in eco-foods. Furthermore, extensive studies must be conducted on people with various chronic diseases to evaluate the health benefits of organic food and investigate the possibility of affording their daily consumption.
